# Chromatin Interaction Analysis with Updated ChIA-PET Tool (V3)

**DOI:** 10.3390/genes10070554

**Published:** 2019-07-22

**Authors:** Guoliang Li, Tongkai Sun, Huidan Chang, Liuyang Cai, Ping Hong, Qiangwei Zhou

**Affiliations:** National Key Laboratory of Crop Genetic Improvement, Agricultural Bioinformatics Key Laboratory of Hubei Province, Hubei Engineering Technology Research Center of Agricultural Big Data, College of Informatics, Huazhong Agricultural University, No. 1, Shizishan Street, Hongshan District, Wuhan 430070, China

**Keywords:** chromatin interactions, genome-wide, high-throughput, chromatin interaction analysis with paired-end tag data analysis, ChIA-PET

## Abstract

Understanding chromatin interactions is important because they create chromosome conformation and link the cis- and trans- regulatory elements to their target genes for transcriptional regulation. Chromatin Interaction Analysis with Paired-End Tag (ChIA-PET) sequencing is a genome-wide high-throughput technology that detects chromatin interactions associated with a specific protein of interest. We developed ChIA-PET Tool for ChIA-PET data analysis in 2010. Here, we present the updated version of ChIA-PET Tool (V3) as a computational package to process the next-generation sequence data generated from ChIA-PET experiments. It processes short-read and long-read ChIA-PET data with multithreading and generates statistics of results in an HTML file. In this paper, we provide a detailed demonstration of the design of ChIA-PET Tool V3 and how to install it and analyze RNA polymerase II (RNAPII) ChIA-PET data from human K562 cells with it. We compared our tool with existing tools, including ChiaSig, MICC, Mango and ChIA-PET2, by using the same public data set in the same computer. Most peaks detected by the ChIA-PET Tool V3 overlap with those of other tools. There is higher enrichment for significant chromatin interactions from ChIA-PET Tool V3 in aggregate peak analysis (APA) plots. The ChIA-PET Tool V3 is publicly available at GitHub.

## 1. Introduction

In eukaryotic cells, the genomes are packaged into the micron-sized nucleus with chromatin as the basic unit. Such packing of genomes into a three-dimensional structure with chromatin interactions is important for DNA replication, DNA damage repair, gene transcription, and other biological functions. Due to the significance of chromatin interactions in biology in general and transcription regulation in particular, the National Institute of Health (NIH) initiated the NIH Common Fund: 4D Nucleome (4DN) Program to study the three-dimensional (3D) genome structure and its spatiotemporal dynamics. Chromatin interaction study is an essential part of the 4DN program, and the Chromatin Interaction Analysis with Paired-End Tag (ChIA-PET) sequencing method [1] is one of the high-throughput methods for generating chromatin interaction data with next-generation sequencing technology.

The ChIA-PET method is a derivative of the Chromosome Conformation Capture (3C) method [2] and is a genome-wide, high-throughput, high-resolution method used to detect chromatin interactions associated with a specific protein of interest, and has been used in a number of applications [3]. The ChIA-PET experiment is based on the idea that the proximal DNA fragments from the same cross-linked molecular complexes can be ligated together [4], and it comprises a few basic steps (Figure 1): cross-linking of the molecules inside the nucleus, shearing the chromatin, precipitating molecules with some antibody of a protein of interest to pull down DNAs associated with the protein, dividing the samples into two aliquots and adding different linkers to two aliquots for linker ligation, combining the two aliquots for proximal ligation of DNAs from the individual molecules, de-cross-linking the proteins from DNAs, digesting the DNAs with the enzyme MmeI, Polymerase Chain Reaction (PCR) amplification, and purifying the final sample for next-generation sequencing. The DNA constructs can be sequenced with the high-throughput DNA sequencing facilities in paired-end mode to generate tens to hundreds of millions of 2 × 36 bp paired-end tag (PET) sequences, or can be sequenced as single reads with a read length of more than 78 bp.

In 2015, Tang et al. improved the original method by developing long-read ChIA-PET [5]. The long reads facilitate higher mapping confidence and base pair coverage. The long-read ChIA-PET experiment includes the following steps (Figure 2): first, millions of cells were fixed in PBS buffer. Next, formaldehyde was added to cross-link the cells and then neutralized. The cross-linked cells were lysed by cell lysis buffer and nuclear lysis buffer. Chromatin was subjected to fragmentation with an average length of 300 bp by sonication. The specific antibody was used to enrich chromatin fragments. After performing the end-repair and A-tailing using T4 DNA polymerase and Klenow enzyme, the Chromatin Immunoprecipitation (ChIP) DNA ends were proximity-ligated by the single biotinylated bridge-linker with the 3’ nucleotide T over-hanging on both strands. Proximity ligation DNA was reverse cross-linked and fragmented, and sequencing adaptors were added simultaneously by using Tn5 transposase. Deoxyribonucleic fragments contained in the bridge linker at ligation junctions were captured by Streptividin beads and used as templates for PCR amplification. These DNA products were subjected to paired-end sequencing (2 × 150 bp) using Illumina Hi-Seq 2500.

The ChIA-PET Tool V3 is a computational package for processing the DNA sequence data generated from the ChIA-PET experimental method, and it consists of 7 steps (Figure 3): (1) linker filtering, (2) mapping the paired-end reads to a reference genome, (3) purifying the mapped reads, (4) dividing the mapped reads into different categories, (5) peak calling from self-ligation PETs, (6) interaction calling from inter-ligation PETs, and (7) visualizing the results. The original pipeline was published in the journal *Genome Biology* in 2010 [4], which was designed to process the data from the 2 × 36 bp paired-end sequencing mode. With the advance in sequencing technology, the cost of sequencing is reduced, and the amount of data increases at the same time. In the long-read ChIA-PET, the length of the paired-end tags is up to 2 × 250 bp [6]. Therefore, we updated ChIA-PET Tool substantially, which included integrating the pipeline of processing short-read data and long-read data, rewriting the shell script of the processing pipeline with Java, revising the linker filtering with multithreading, adjusting the step of mapping to reduce running time, generating the statistics of the result, and evaluating the quality of the data. In this paper, we demonstrate how to apply the ChIA-PET Tool V3 to the public ChIA-PET data and illustrate the details and interpretation of the results to facilitate the usage of ChIA-PET Tool V3.

## 2. Materials and Methods

### 2.1. Equipment

#### 2.1.1. Hardware

Our example of short-read ChIA-PET data was run on CentOS release 7.3.1611 with Intel(R) Xeon(R) CPU E5-2630 0 @ 2.30GHz.

Our example of long-read ChIA-PET data was run on CentOS release 6.6 with Intel(R) Xeon(R) CPU E5-2620 v3 @ 2.40GHz.

#### 2.1.2. Required Supporting Software

Java is a popular platform-independent programming language and can be run on any machines with a Java Virtual Machine (JVM); BWA [7,8] is used to map ChIA-PET sequencing reads to a reference genome; SAMtools [9] is used to convert the alignment output from SAM format to BAM format; BEDTools [10] is required to convert the files from BAM format to BED format; R environment [11] and its packages are used to compute the *p*-values in peak calling and interaction calling and generate the graphs for visualization.

Download the supporting software listed as follows:(1)Java Development Kit (JDK) ≥ 1.8 [12](2)BWA [8](3)SAMtools [9](4)BEDTools [10](5)R [11](6)R package grid (install.packages(“grid”))(7)R package xtable (install.packages(“xtable”))(8)R package RCircos (install.packages(“RCircos”))

Download the file ChIA-PET Tool V3.zip from ChIA-PET Tool V3 package on GitHub (https://github.com/GuoliangLi-HZAU/ChIA-PET_Tool_V3) and unpack it.

#### 2.1.3. Required Supporting Data

To run ChIA-PET Tool V3, the linker sequence, the genome sequence and its mapping index, the lengths of the individual chromosomes, and cytoband data of the interested genome are required. In this paper, human reference genome hg19 [13], chromosome size data [14] and cytoband data [15] were downloaded from the University of California, Santa Cruz (UCSC) website. The random sequences in the genome were removed before further processing. Then, BWA was used to build a genome index for mapping sequence reads.

#### 2.1.4. Example Chromatin Interaction Analysis with Paired-End Tag (ChIA-PET) Data

For short-read test data, ChIA-PET data associated with RNA polymerase II (RNAPII) from human K562 cells was downloaded from NCBI GEO with accession numbers GSM832464 and GSM832465 and dumped to FASTQ format. The raw reads from these two replicates were combined for further processing.

For long-read test data, we use CCCTC binding factor (CTCF) ChIA-PET data from human GM12878 cells with NCBI GEO accession number GSM1872886.

### 2.2. Updates

#### 2.2.1. Programming Language of Processing Pipeline

In ChIA-PET Tool V3, we changed the programming language of processing pipeline and used Java program instead of shell script in the previous version. Then, we compiled and packaged the program into jar format. This unified the programming language and better implemented the encapsulation. However, we used other software in our tool by executing shell scripts. For example, we used BWA when mapping to a reference genome. Thus, these scripts were retained, which could be generated by the program according to the corresponding input options.

#### 2.2.2. Integrating the Short-Read and Long-Read ChIA-PET Pipeline

Previously, we used two pipelines to process short-read and long-read ChIA-PET data, respectively. In fact, there are many of the same processes in the two pipelines. The main differences between them include the stages of linker filtering and mapping to the reference genome. Therefore, we can choose to execute the processes of short-read pipeline or long-read pipeline by setting one option in the current pipeline.

#### 2.2.3. Multithreading in Linker Filtering

In order to speed up the pipeline, ChIA-PET Tool V3 uses multithreading in the linker filtering stage. With the Java interface BlockingQueue, one thread reads thousands of PETs from two FastQ files as a data block and puts it into the queue. Then, multiple threads get data blocks from the queue and process them.

#### 2.2.4. Mapping to the Reference Genome with Burrows-Wheeler Aligner (BWA)

In the original version of ChIA-PET Tool, there were no satisfactory mapping tools available at that time. We used a home-made mapping method, BatMan, for mapping the reads to the reference genome, which was later published as BatMis [16]. In the current version, we use the popular mapping tool, BWA [7,8], for mapping.

#### 2.2.5. Option for Chromatin Interaction Calling without Peaks as Inputs

In the current pipeline, we provide an option for chromatin interaction calling without peaks as inputs at this stage. The reason for this is that some interactions may not be supported by strong peaks at the anchor regions.

#### 2.2.6. Statistics Report for Different Steps in the Data Processing

The statistics for the data processing procedure are summarized in an HTML file for users to check the data. The chromosome interaction maps contain intra- and inter-chromosomal interaction information. When there are tens of thousands of interactions, some chromatin interactions cannot be visualized clearly in chromosome scale. We use Discuz [17] to achieve image zooming. Discuz is an image processing toolkit based on JavaScript. When we click on an image, the image will be displayed in a new window, which can be dragged and zoomed by mouse.

### 2.3. Procedure

The ChIA-PET Tool V3 is an easy-to-use pipeline, and we can simply run it using one command line with some options (Appendix A). Users must set the 10 necessary options, whereas other options have default values. In particular, the directories of data should be set properly to make sure that the programs can run smoothly. The ChIA-PET Tool V3 will create a folder named “OUTPUT_PREFIX” in the “OUTPUT_DIRECTORY”. The default value of “OUTPUT_DIRECTORY” is the master folder “ChIA-PET_Tool_V3/”, and the default value of “OUTPUT_PREFIX” is “out”.

The ChIA-PET Tool V3 includes 7 steps for data processing (Figure 3). We can use the option “start_step” to select which step to start with. The following sections illustrate the main steps in the ChIA-PET Tool V3.

#### 2.3.1. Linker Filtering

The ChIA-PET Tool V3 could process short-read or long-read ChIA-PET data by setting option “mode”. Users are required to provide the linker sequences in a linker file. For short-read data, the linker sequences are two half-linker sequences. For long-read data, the linker sequences are from +/− strands of a bridge linker. When the length of the linker sequences is changed, the option “minimum_linker_alignment_score” should be changed accordingly. If users decide to process short-read data, the start position of the barcode and the length of the barcode would be calculated in the program according to the linker sequences. If users decide to process long-read data, there is no barcode information for the linker sequence.

According to the design of the short-read ChIA-PET experiment, two half-linkers would be combined into a full linker after proximal ligation. After MmeI digestion, we can get the DNA constructs in a “tag-linker-tag” format (Figure 4). The constructs are 78 bp, consisting of two tags of 20 bp from interacting DNA fragments and one full linker of 38 bp. With the 2 × 36 bp paired-end sequencing mode, reads of 36 bp will be sequenced from each end of the constructs, which contains 20 bp from the DNA fragments and 16 bp from the half-linker in the ideal condition.

In the stage of linker filtering, a local sequence alignment algorithm is used to align the designed linker sequences to the part of linker on the real read sequences. If the alignment score is higher than a user-defined threshold, the linker is identified. In the original design, the barcodes in the linker sequences are 2 bp (AT or CG) and are not enough to differentiate the half-linkers A and B in some cases. Now the barcodes are changed to 4 bp (TAAG or ATGT) [18] in order to improve the differentiation of barcodes. Both reads in one PET aligned to the same linker are named “same-linker PET”. If one read is aligned to linker A and another read is aligned to linker B, this PET is named “different-linker PET”. If only one read is aligned to linker A or linker B, or neither read is aligned to linker A and linker B, this PET is named “ambiguous-linker PET”. Then, the part of linker on the read sequences is trimmed and the remaining part is kept for further analysis, which should be at least 18 bp long. We only use the same-linker PETs for further analysis.

According to the design of the long-read ChIA-PET experiment, we use a bridge linker for proximal ligation and Tn5 transposase to fragment DNA. Since the location of the interruption is random, the length of DNA fragments is not fixed. In the linker filtering stage, we select the PET where one read is aligned to the plus (+) strand of linker, whereas another read is aligned to the minus (−) strand of the linker for further analysis.

#### 2.3.2. Mapping to a Reference Genome

After linker filtering, the trimmed paired DNA fragments are mapped to a reference genome. A Burrows–Wheeler-transform-based method, BatMis [16], was used to generate customized output in SAM format in the previous version. In the current version, we used BWA for mapping. We assume that the genome index is already built for BWA, and the path and the prefix of the index files are specified with variable “GENOME_INDEX”.

For short-read data, the DNA sequences after linker filtering are around 20 bp, option “aln” in BWA is used for reads mapping and option “samse” is used to convert the mapping results into SAM format. After mapping, we use a Java program to extract the uniquely mapped reads with high mapping quality score and generate a BEDPE file with multithreading.

For long-read data, considering the difference in the length of DNA sequences after linker filtering, option “aln” is used for PETs with a length of less than 55 bp, and option “mem” is used for PETs with a length of more than 55 bp, so that the accuracy of mapping to the reference genome can be improved. After extracting the uniquely mapped reads, SAMtools is used to convert the files from SAM format to BAM format, and BEDTools is used to convert the files from BAM format to BEDPE format.

#### 2.3.3. Cleaning the Mapped Paired-End Tags (PETs)

Different sources of noise exist in ChIA-PET data, including duplicated reads from PCR amplification, variable cutting length from MmeI enzyme, and sequencing errors due to the repetitive sequences in the genome, which are considered separately in the data purification stage. Firstly, the reads that are exactly mapped to the same location (Figure 5A) are likely to be the PCR duplicates, and only one of them is kept for further processing. Secondly, if different PETs have both tags within 2 bp at each side (Figure 5B), there is a high chance that they are from the same DNA fragment with variable MmeI cutting length or linker filtering cutting. Such PETs are combined as one PET for further processing.

This stage consists of two main operations to remove duplicate PETs from amplification and other noises: (1) Merge all PETs with the same mapping locations, probably due to PCR amplification, into one unique PET; (2) merge all the similar PETs (within ±2 bp at the both ends of different PETs) into one unique PET.

#### 2.3.4. Categorization of the Paired-End Tags

By the ChIA-PET experiment design, the two tags in each paired-end read could come from a single DNA fragment or two different DNA fragments by ligation. In order to identify the chromatin interactions, we should use the PETs from different DNA fragments. We divide the PETs into different categories (Figure 6): self-ligation PETs (PETs from the single DNA fragments), intra-chromosomal inter-ligation PETs (PETs from two different DNA fragments in the same chromosome), inter-chromosomal inter-ligation PETs (PETs from DNA fragments in two different chromosomes) and others.

To separate the self-ligation PETs from intra-chromosomal inter-ligation PETs, we need to determine the genomic distance cutoff between the two tags of the PETs in the same chromosome. By the experiment design and sequencing protocol, the self-ligation PETs are (1) from minus-plus (−/+) chromosome strand composition, (2) with short genomic span, and (3) the tag from minus strand should have smaller genomic coordinate. The intra-chromosomal inter-ligation PETs are from all possible strand compositions. By comparing the genomic span distributions for PETs with different strand compositions, it will give a clue of the span threshold for self-ligations. Figure 7 shows that the genomic spans from +/+, +/−, and −/− strand compositions have similar distributions, whereas genomic spans from −/+ strand composition have different distributions—there are much more PETs with genomic spans less than 10 kb from −/+ strand composition. Based on the assumption that the intra-chromosomal inter-ligation PETs should be similarly distributed among different strand compositions, the extra PETs with genomic span less than 10 kb from −/+ strand composition are from self-ligation. Then, the difference of genomic span distributions from −/+ strand composition and the average distribution from other strand compositions is used to determine the span cutoff for self-ligation. The log-log plot of genomic span distribution difference in Figure 8 shows that the cutoff is around 8 kb. For the illustration purpose, we use 8 kb as the default cutoff to classify the PETs into different categories.

#### 2.3.5. Peak Calling

Peak calling with self-ligation PETs are similar to the peak calling from ChIP-Seq data with paired-end sequencing mode. The enrichment of the PETs in a genomic region is considered as the potential binding sites of the protein of interest. In ChIA-PET Tool V3, overlapping regions of self-ligation PETs are used to define transcription factor binding sites and Poisson distribution similar to MACS [19] is used to calculate the *p*-values for the peaks.

#### 2.3.6. Interaction Calling

Overlapping regions of inter-ligation PETs are used to define interacting regions and hyper-geometric distribution is used to calculate *p*-value for the interactions. Chromatin interaction calling is based on the overlapping of the extended tags from different PETs. The extension of the tags is based on the fact that the sequencing reads are just part of DNA fragments in the experiment. The tag extension length is determined by the median span of the self-ligation PETs, which is around 500 bp. For illustration purpose, we use 500 bp as the default parameter. The statistical significance of the interaction is assessed by *p*-value from a hyper-geometric model and adjusted with the Benjamini–Hochberg method for multiple hypothesis testing. We used adjusted *p*-value cutoff 0.05 to get significant interactions (false discovery rate, FDR, < 0.05).

In the current pipeline, the interaction calling does not depend on the given peaks, as some interactions may not be supported by strong peaks at the anchor regions. This is why we changed the interaction calling to the overlap of the extended tags from the PETs. Still, we provide the option for users to call chromatin interactions with any given regions, which can be from ChIP-Seq peak calling, or any genomic regions of interest, such as the promoter regions.

#### 2.3.7. Statistics Report and Visualization

The results of ChIA-PET data analysis are visualized in two ways: (1) the statistics of the data quality and (2) the list of peaks and interactions. During the execution of ChIA-PET Tool V3, the statistics of the library is generated and summarized in an HTML file. The ChIA-PET data (original mapping PETs, peaks and interactions) are converted into BED format for visualization with a genomic browser, such as a UCSC browser.

## 3. Results

### 3.1. Anticipated Results

The ChIA-PET Tool V3 can process the next-generation sequence data from the ChIA-PET experiment to generate enriched binding peaks of the protein of interest and the related chromatin interactions. We demonstrated the application of ChIA-PET Tool V3 with RNAPII-associated ChIA-PET data from human K562 cells as an example.

The results from linker filtering include a few summary statistics. Figure A1A shows the distribution of the best linker alignment scores from the designed linker sequences to the reads. We can see that most of the alignment scores are 10, which means that most of the linker sequences in the reads are 10 base pairs. Figure A1B shows the distribution of linker alignment score differences between the best-aligned linker and the second-best aligned linker. This distribution is used to check the difference between the best-aligned linker and the second-best aligned linker, in case there are ambiguities to locate the linker in the reads. Figure A1C shows the distribution of tag lengths after trimming the best-aligned linker sequences from the reads. We can see that most of the tag lengths are 20 or 21 bp, which is consistent with enzyme MmeI’s digestion property. Table 1 reports the proportion of each linker combination in the reads. The results show that most (90.73%) of the PETs are composed of the same-linkers (A_A or B_B). This indicates proper proximity-ligation within individual molecules.

Figure 9 shows a screenshot of binding profile and chromatin interactions from the example data set. Peaks from self-ligation PETs of RNAPII-associated ChIA-PET are mainly enriched around gene promoter regions. The interactions are mainly between gene promoter regions and other regulatory elements, which indicate that RNAPII-associated chromatin interactions are involved in gene transcription regulation. Table A1 shows the first six lines of the peak output file from Step 5-peak calling. Table A2 shows the first 6 lines of the interaction output file from Step 6-interaction calling. Table 2 and Table 3 summarize the statistics of the clusters. We can see that the majority of the clusters are intra-chromosomal interactions, and most of the interactions span within 1Mb. Figure A2 shows the binding peaks and intra-chromosomal chromatin interactions in the chromosome scale. The binding peaks are distributed all over the whole genome. Most of the intra-chromosomal chromatin interactions are within a short span (within 1 Mb) and a small proportion of chromatin interactions can span a very long distance. Figure A3 shows the inter-chromosomal chromatin interactions in the circular view. Compared to intra-chromosomal chromatin interactions, there are fewer inter-chromosomal chromatin interactions.

### 3.2. Evaluation of Data Quality

One concern for the users is the quality of the ChIA-PET libraries they generated or used. For evaluation of the data, we provide the summary statistics of the results from the ChIA-PET libraries.

#### 3.2.1. Results of Short-Read ChIA-PET Data

Table 4 shows the statistics of results after finishing the data analysis. Table 5 shows the statistics of ChIA-PET libraries. The 1st line in Table 5 is the percentage of the same-linker PETs over the total PETs. Due to the nature of the ChIA-PET design, there will be more same-linker PETs than different-linker PETs. In general, we see the percentage of the same-linker PETs varying from 60% to 99%. If such a percentage is above 75%, the library is good in the linker composition level. The second line is the percentage of the uniquely mappable PETs over the total PETs, which varies with the libraries. The 3rd line is the percentage of the PETs after merging those mapped to the same positions exactly over the uniquely mappable PETs. If this percentage is too low (less than 30%), it means that there are more PETs from PCR amplification, and the data are already near saturated. Then, there is no point in re-sequencing the library for more distinct PETs. If the PETs which, after merging those mapped to the same positions exactly, are 70% or more of the uniquely mappable PETs, deeper sequencing can be applied to get more data for the library. The 4th line is the percentage of inter-ligation PETs over all the PETs after purification. This is about the efficiency catching the interacting PETs. If this percentage is low, it means that the library has too few inter-ligation PETs and is not good enough for chromatin interaction detection. The 5th line is the percentage of intra-chromosomal inter-ligation PETs over the total inter-ligation PETs. From the current understanding, most of the chromatin interactions are within the individual chromosomes. Therefore, there should be more intra-chromosomal PETs than inter-chromosomal PETs. If there are more inter-chromosomal PETs, it means that the proximity ligation introduces many random ligations. The 6th line is the peak number. This depends on the transcription factor used, and should be compared with the background knowledge or available ChIP-Seq data. For RNAPII and CTCF, there are tens of thousands of peaks in a good ChIA-PET library from human and mouse. The 7th line is the number of chromatin interactions. This depends on the transcription factors used. For RNAPII and CTCF, there are tens of thousands of interactions in a good ChIA-PET library from human and mouse.

#### 3.2.2. Results of Long-Read ChIA-PET Data

The statistical indicators of results in the long-read ChIA-PET data are similar to those in the short-read analysis results (Table 6 and Table 7).

### 3.3. Comparing Results with Other Tools

#### 3.3.1. Comparing Results of Short-Read Data

In recent years, with the development of the ChIA-PET method, there are more and more published tools, such as Mango [20], ChIA-PET2 [21,22] and ChiaSig [23], for processing and analyzing ChIA-PET data. In order to evaluate the ChIA-PET Tool V3, we used ChIA-PET data associated with RNAPII from human K562 cells as input data and compared the results of the ChIA-PET Tool V3 with other tools (Table 8). When mapping PETs to a reference genome, Mango used Bowtie, whereas ChIA-PET Tool V3 and ChIA-PET2 used BWA. Meanwhile, the standard of the mapping quality score was different. We selected 30 as the cutoff of the quality score in BWA and selected 40 as the cutoff of the quality score in Bowtie. ChiaSig only performed the stage of significant loop calling. We used a PET cluster file of the ChIA-PET Tool V3 as input for ChiaSig. For significant interactions in Table 8, there are at least three supportive PETs in one interaction. The cutoff of the FDR is 0.05. Then, we calculated the distribution of PET counts in significant interactions (Figure A4). Most interactions have three PETs in the ChIA-PET Tool V3.

In Table 8, Mango could get more non-duplicate PETs after mapping to the human genome. This is because of the difference between Bowtie and BWA. Both ChIA-PET2 and ChIA-PET Tool V3 used BWA in the mapping stage and selected 30 as the cutoff of quality score. Mango used Bowtie to map the reads. When we replaced Bowtie with BWA in Mango, we got the number of non-duplicate PETs in Mango as 7,286,203, which is similar to ChIA-PET2 and ChIA-PET Tool V3. Mango and ChIA-PET2 can detect more peaks, whereas ChIA-PET Tool V3 can detect more significant interactions. The overlaps of peaks and significant interactions detected by different tools are shown in Figure 10. More than 92% peaks of ChIA-PET Tool V3 overlap with Mango and ChIA-PET2. Scatter plots (Figure 11) and box plots (Figure 12) represent the peak intensity between each two different tools. The Pearson correlation coefficients in scatter plots between the three tools are about 0.9. In particular, the Pearson correlation coefficient is 0.94 between ChIA-PET Tool V3 and ChIA-PET2.

For significant interactions in Figure 10, we used interactions with at least three supportive PETs and an FDR of less than 0.05. Most of interactions of other tools overlap with ChIA-PET Tool V3. A total of 871 interactions (blue area) of ChIA-PET Tool V3 have no overlap with other tools. Among them, 358 interactions are not supported by peaks. However, all 1984 interactions (purple area) of ChIA-PET2 which have no overlap with other tools are supported by peaks. We compared the genomic proximity of the subset interactions unique to one tool (interactions from one tool without overlap with interactions from other tools, such as 871 for ChIA-PET Tool V3, 512 for Mango, and 1984 for ChIA-PET2), as shown in Figure 13. Most genomic distances from interactions unique to ChIA-PET Tool V3 are around 10 kb, whereas interaction distances unique to Mango and ChIA-PET2 are around 100 kb to 1 Mb. We compared the interactions of ChIA-PET Tool V3, which are overlapped with those from ChiaSig and the interactions which are not overlapped with those from ChiaSig on PET count, genomic distance between two anchors, and −log(*p*-value). The results in Figure 14 show that the distributions of PET counts, genomic distance, and −log10(*p*-value) are similar.

Box plots (Figure 15) and scatter plots (Figure 16) represent the interaction intensity between each two different tools. The difference of common and unique parts in the box plots is significant except for ChIA-PET Tool V3 and ChiaSig. The Pearson correlation coefficients between ChIA-PET Tool V3 and other tools are high. As we used PET cluster file generated from ChIA-PET Tool V3 as an input of ChiaSig, all the significant interactions detected by ChiaSig overlap with interactions from ChIA-PET Tool V3. In Mango and ChIA-PET2, they used a method for chromatin interaction calling with peak regions. However, ChIA-PET Tool V3 used the option for chromatin interaction calling without peaks as input, as some interactions may not have peaks at the anchor regions.

We generated aggregate peak analysis (APA) plots [24] (Figure 17) from the ChIA-PET data for significant chromatin interactions with at least 3 supportive PETs and FDR of less than 0.05 detected by different tools. To generate APA plots, we used BEDPE files from the ChIA-PET data after removing duplicates from different tools to build interaction matrices [25]. Then, we summed interaction counts for pairs of loci in 25 kb bins with Juicer tools [26]. The level of different sets of interactions can be quantified by an APA score. Higher scores indicate higher enrichment of interactions. In these plots, APA scores for ChIA-PET Tool V3 and ChiaSig are similar and both are higher than Mango and ChIA-PET2.

#### 3.3.2. Comparing the Results of Long-Read Data

Considering that both ChIA-PET Tool V3 and ChIA-PET2 can process and analyze long-read ChIA-PET data, we executed ChIA-PET Tool V3 and ChIA-PET2 using CTCF ChIA-PET data from human GM12878 cells as an input. In Table 9, one interaction has at least three PETs and FDR of less than 0.05. In the long-read ChIA-PET experiment, we can get DNA sequences with different lengths after proximity ligation (Figure A5). Maybe we can find the linker on one end but the tag length of another end is too large, and we could not find the linker by sequencing. The ChIA-PET Tool V3 could get more non-chimeric PETs because we used different linker combinations, such as AB, BA, AX, XA, BX, and XB PETs, whereas ChIA-PET2 only used AB PETs and BA PETs. Here, X means that no linker was found in the reads.

Figure A6 is the PET count distribution of significant interactions. Most interactions detected by ChIA-PET Tool V3 have three PETs. We compared the results and generated overlap plots of peaks and significant interactions detected by ChIA-PET Tool V3 and ChIA-PET2 (Figure 18). Ninety-nine percent of the peaks of ChIA-PET Tool V3 overlap with ChIA-PET2. For significant interactions in Figure 16B, we used interactions with at least 3 supportive PETs and FDR of less than 0.05. The ChIA-PET Tool V3 could find more significant interactions. Further, 94% of the significant interactions of ChIA-PET2 overlap with those of ChIA-PET Tool V3. A total of 382,396 interactions of the ChIA-PET Tool V3 have no overlap with other tools. Among them, 205,395 interactions are not supported by peaks. However, all of the 2288 interactions of ChIA-PET2 which have no overlap with ChIA-PET Tool V3 are supported by peaks.

In the scatter plot of peaks (Figure 19A), the Pearson correlation coefficient between ChIA-PET Tool V3 and ChIA-PET2 is 0.94. The difference of common and unique parts in the box plot (Figure 19B) is significant for ChIA-PET Tool V3. In the scatter plot of significant interaction (Figure 19C), the Pearson correlation coefficient between ChIA-PET Tool V3 and ChIA-PET2 is 0.79. The difference of common and unique parts in the box plot (Figure 19D) is significant for ChIA-PET Tool V3. Then, we generated APA plots with ChIA-PET data for chromatin interactions with at least three supportive PETs in an interaction and FDR of less than 0.05, as shown in Figure 20. The APA scores of ChIA-PET Tool V3 and ChIA-PET2 are similar.

## 4. Discussion

In this paper, we introduced the design and usage of ChIA-PET Tool V3. By processing short-read RNAPII and long-read CTCF ChIA-PET data, we demonstrated how to apply ChIA-PET Tool V3 to the public ChIA-PET data and provided details of the results.

The ChIA-PET Tool V3 can process short-read and long-read ChIA-PET data from paired-end reads. It would generate enriched binding peaks and chromatin interactions associated with a protein of interest. Multiple log files and statistics are generated for tracing the processing issues if any part of the process goes wrong and for evaluating the quality of the libraries. The ChIA-PET Tool V3 has the advantages of user-friendliness, multithreading, and result visualization. During the execution of ChIA-PET Tool V3, the statistics of the library are generated and summarized in an HTML file. Checking the information of the data in charts is clear and straightforward. In particular, users can zoom in or out of the plots of interactions using the mouse wheel to check the interactions in detail. With the improvement of 3D genome technologies, we believe more researchers need tools like ChIA-PET Tool V3 to analyze chromatin interaction data and obtain more information to understand fundamental aspects and structures of the genome.

## Figures and Tables

**Figure 1 genes-10-00554-f001:**
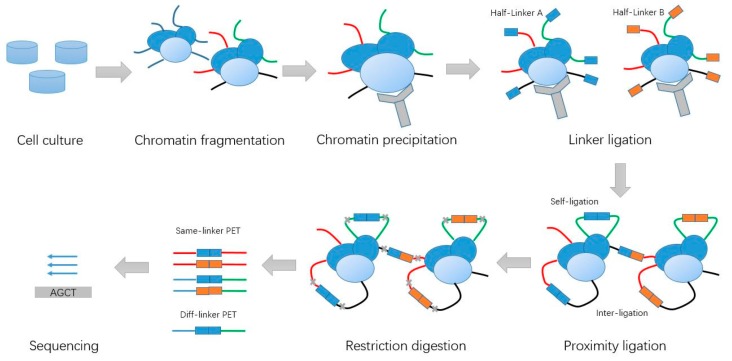
Schematic of Chromatin Interaction Analysis with Paired-End Tag (ChIA-PET) experiment steps.

**Figure 2 genes-10-00554-f002:**
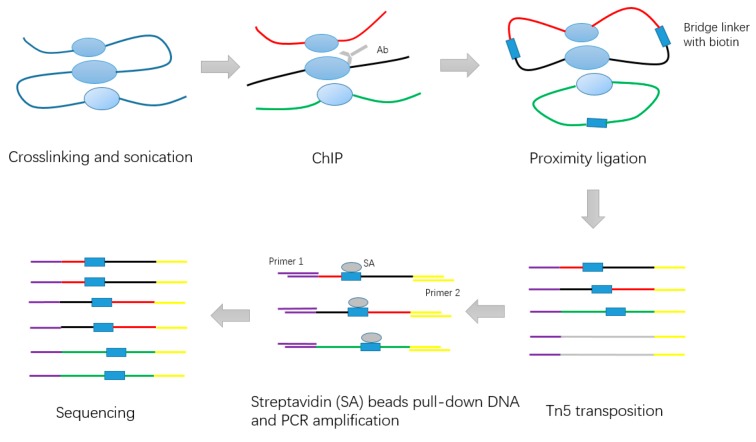
Schematic of long-read ChIA-PET experiment steps.

**Figure 3 genes-10-00554-f003:**
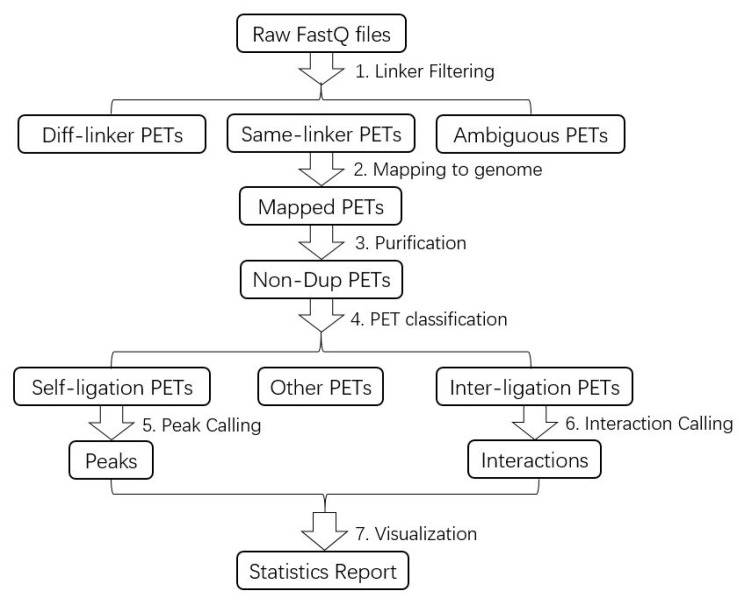
Flowchart of ChIA-PET Tool V3 for ChIA-PET data analysis.

**Figure 4 genes-10-00554-f004:**
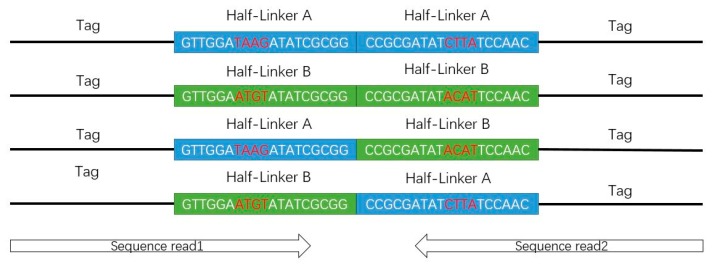
Deoxyribonucleic acid constructs from the experiments in tag-linker-tag format.

**Figure 5 genes-10-00554-f005:**
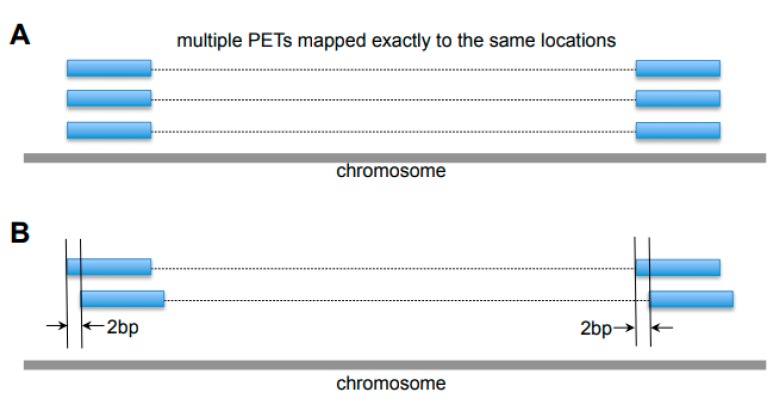
Examples of noises in the ChIA-PET data. (**A**) Duplicates of Paired-End Tags (PETs) from Polymerase Chain Reaction (PCR) amplification, which are mapped exactly to the same locations. (**B**) Different PETs with tags within 2 bp at both ends.

**Figure 6 genes-10-00554-f006:**
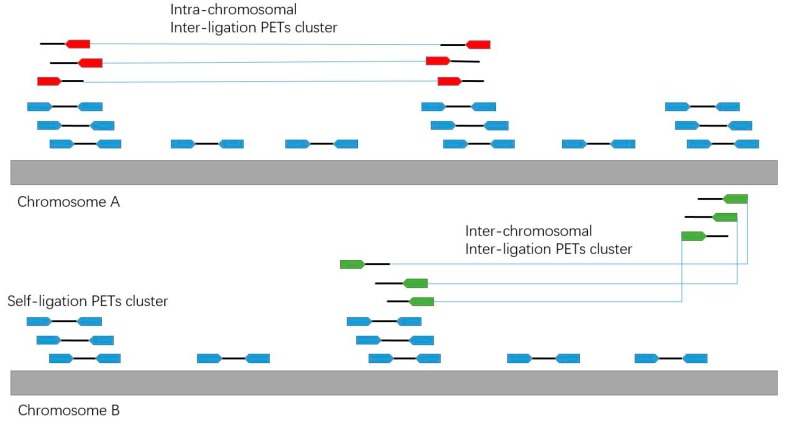
Illustration of the different categories of PETs.

**Figure 7 genes-10-00554-f007:**
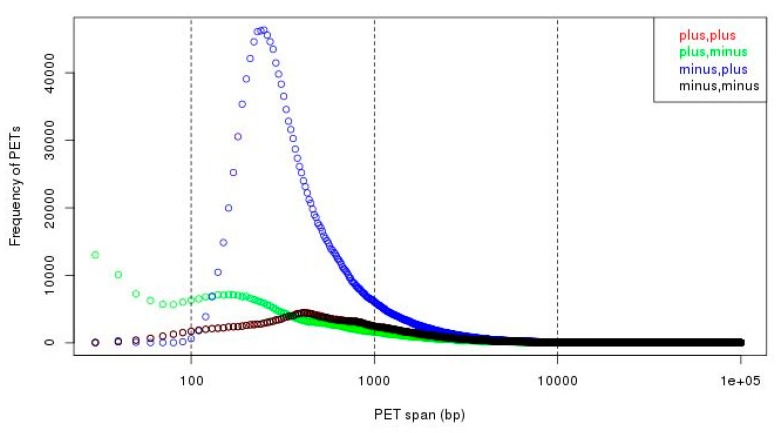
Genomic span distributions from different strand compositions. We can see that there are much more PETs with minus-plus (−/+) strand composition in short span (less than 10 kb, especially less than 1 kb).

**Figure 8 genes-10-00554-f008:**
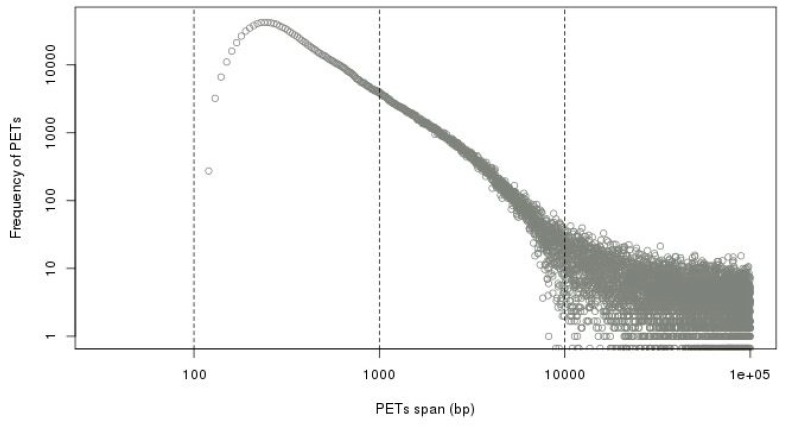
Difference of genomic span distributions from minus-plus (−/+) strand composition and average distribution from other strand compositions in log-log plot. It shows that the self-ligation cutoff is around 8 kb.

**Figure 9 genes-10-00554-f009:**
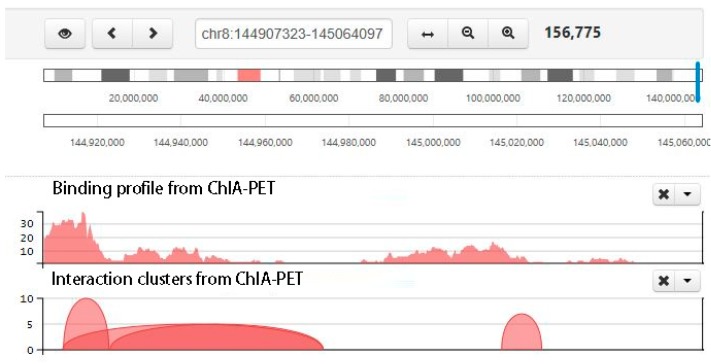
Screenshot of chromatin clusters and protein-binding peaks from ChIA-PET. There are multiple tracks, including interaction clusters and binding profile from ChIA-PET. In the interaction cluster track, two ends of each curve are the interaction anchors, and the height of the curve is the pet count between the interaction anchors. The higher the curve, the stronger the interaction.

**Figure 10 genes-10-00554-f010:**
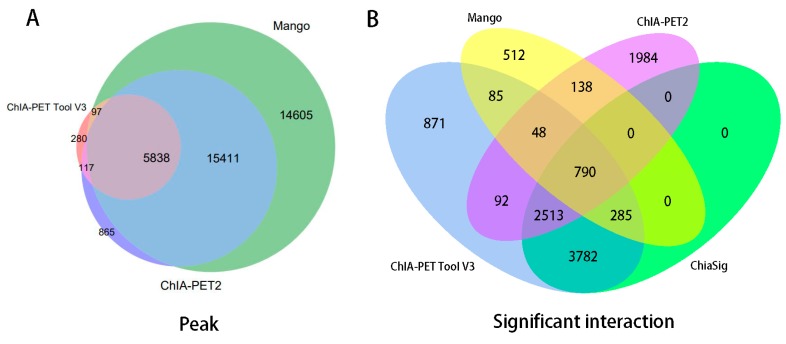
Overlap of (**A**) peaks and (**B**) interactions detected by different tools from human K562 cells. Significant interactions are required for at least three supportive PETs and FDR less than 0.05.

**Figure 11 genes-10-00554-f011:**
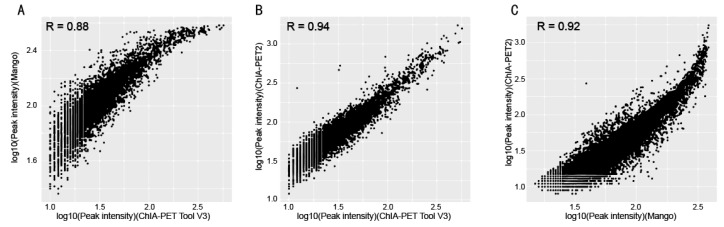
Scatter plots of peak intensity between different tools from human K562 cells. (**A**) Peak intensity between ChIA-PET Tool V3 and Mango. (**B**) Peak intensity between ChIA-PET Tool V3 and ChIA-PET2. (**C**) Peak intensity between Mango and ChIA-PET2.

**Figure 12 genes-10-00554-f012:**
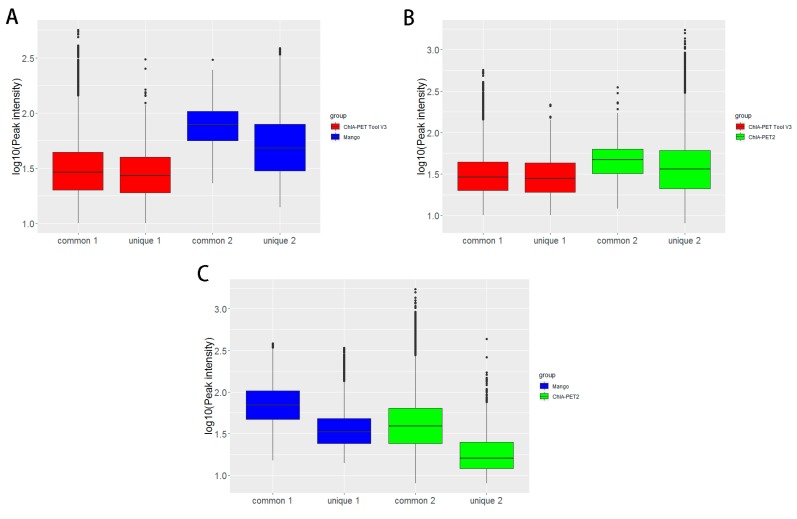
Box plots of peak intensities between different tools from human K562 cells. (**A**) Peak intensity between ChIA-PET Tool V3 and Mango. (**B**) Peak intensity between ChIA-PET Tool V3 and ChIA-PET2. (**C**) Peak intensity between Mango and ChIA-PET2. The common part represents peak intensities with overlap between two tools. The unique part represents peak intensity without overlap between two tools. Middle line denotes median; box denotes Interquartile Range (IQR); whiskers denote 1.5 × IQR.

**Figure 13 genes-10-00554-f013:**
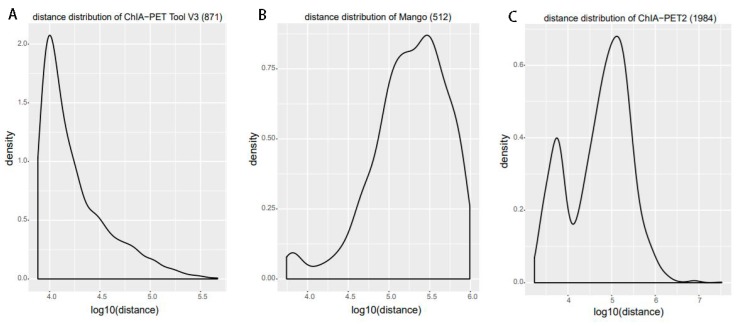
Genomic proximity of the subset interactions unique to each tool. (**A**) Distance distribution of the unique interactions from ChIA-PET Tool V3. (**B**) Distance distribution of the unique interactions from Mango. (**C**) Distance distribution of the unique interactions from ChIA-PET2.

**Figure 14 genes-10-00554-f014:**
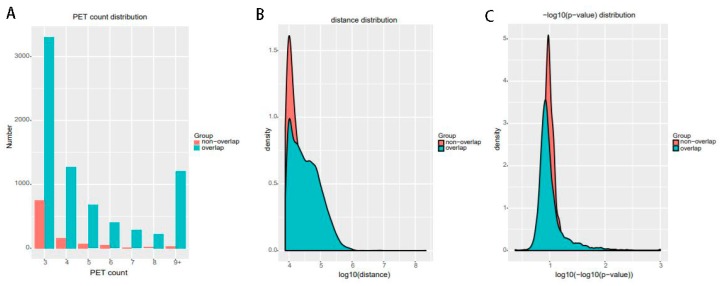
The interactions of ChIA-PET Tool V3 which are overlapped with those from ChiaSig and the interactions which are not overlapped with those from ChiaSig on (**A**) PET count, (**B**) genomic distance between two anchors and (**C**) −log(*p*-value).

**Figure 15 genes-10-00554-f015:**
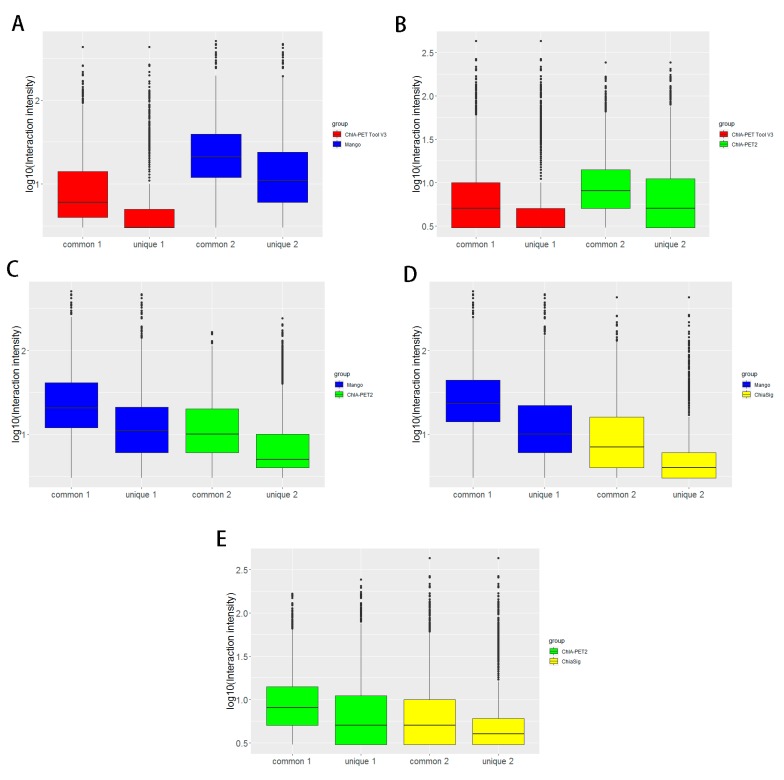
Box plots of interaction intensity between different tools from human K562 cells. (**A**) Interaction intensity between ChIA-PET Tool V3 and Mango. (**B**) Interaction intensity between ChIA-PET Tool V3 and ChIA-PET2. (**C**) Interaction intensity between Mango and ChIA-PET2. (**D**) Interaction intensity between Mango and ChiaSig. (**E**) Interaction intensity between ChIA-PET2 and ChiaSig.

**Figure 16 genes-10-00554-f016:**
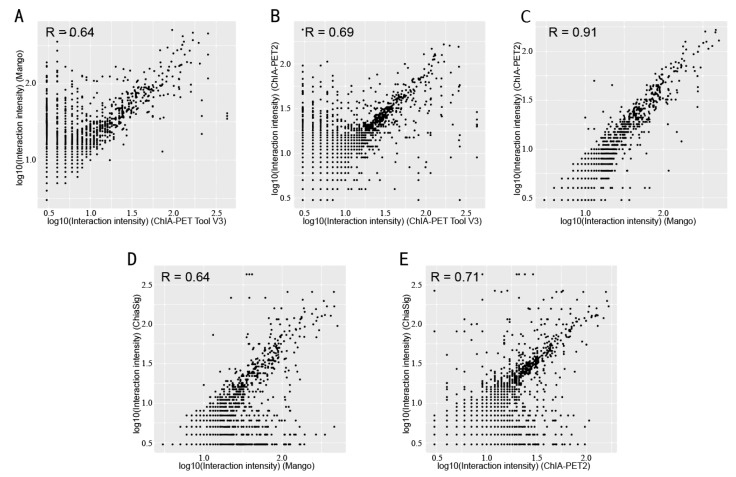
Scatter plots of interaction intensity between different tools from human K562 cells. (**A**) Interaction intensity between ChIA-PET Tool V3 and Mango. (**B**) Interaction intensity between ChIA-PET Tool V3 and ChIA-PET2. (**C**) Interaction intensity between Mango and ChIA-PET2. (**D**) Interaction intensity between Mango and ChiaSig. (**E**) Interaction intensity between ChIA-PET2 and ChiaSig.

**Figure 17 genes-10-00554-f017:**
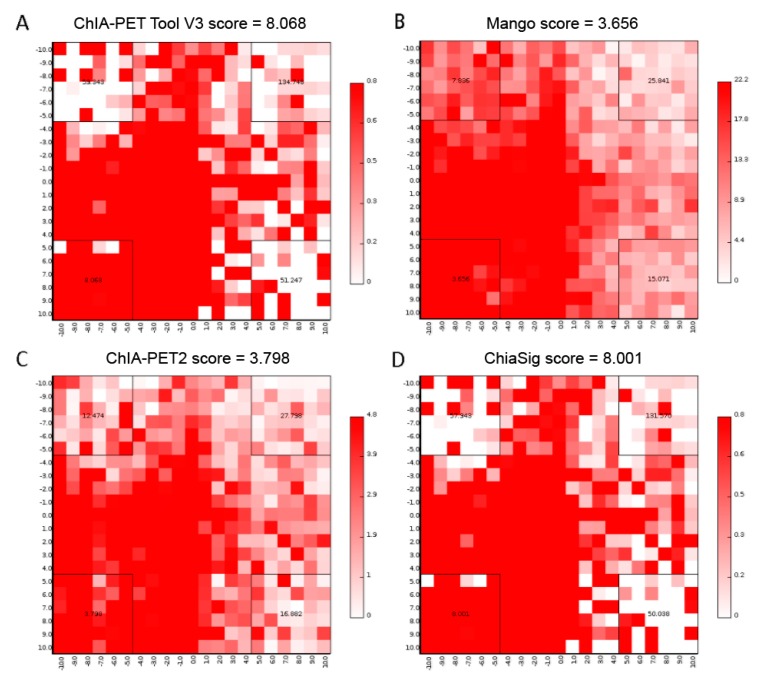
Aggregate Peak Analysis (APA) plots of significant interactions detected by different tools from human K562 cells. (**A**) APA plot of significant interactions detected by ChIA-PET Tool V3. (**B**) APA plot of significant interactions detected by Mango. (**C**) APA plot of significant interactions detected by ChIA-PET2. (**D**) APA plot of significant interactions detected by ChiaSig. Significant interactions are required at least 3 supportive PETs and FDR less than 0.05. Interaction matrices are built with BEDPE files of ChIA-PET data after removing duplicates. Interaction counts are summed for all pairs of loci in 25 kb bins.

**Figure 18 genes-10-00554-f018:**
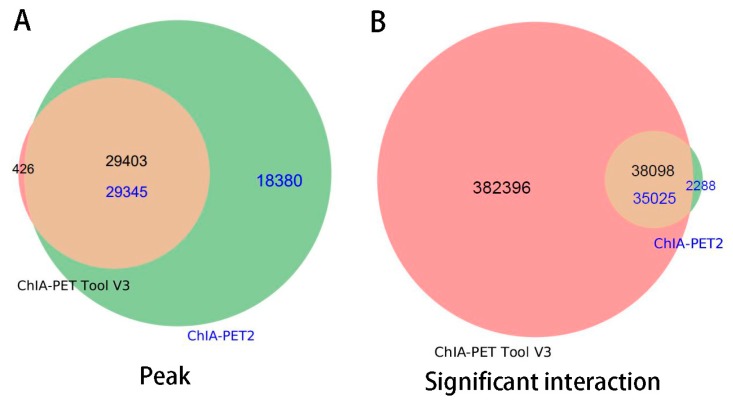
Overlap of peaks and interactions detected by different tools from human GM12878 cells. Significant interactions require at least three supportive PETs and FDR of less than 0.05.

**Figure 19 genes-10-00554-f019:**
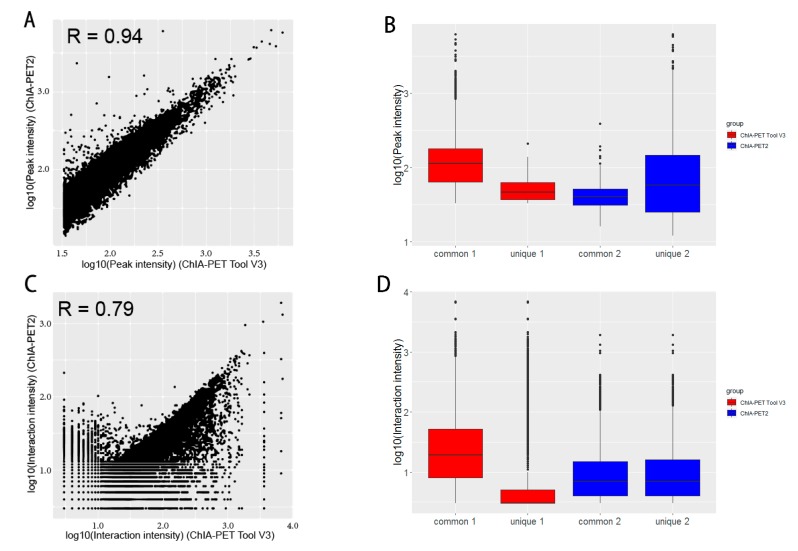
Comparison of peaks and interactions from ChIA-PET Tool V3 and ChIA-PET2 in CTCF ChIA-PET data. (**A**,**B**) Scatter plot and box plot of peak intensity from human GM12878 cells, respectively. (**C**,**D**) Scatter plot and box plot of interaction intensity from human GM12878 cells, respectively.

**Figure 20 genes-10-00554-f020:**
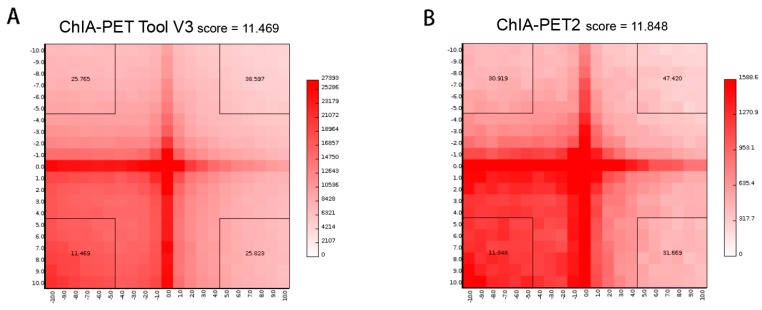
Aggregate peak analysis (APA) plots of significant interactions detected by different tools from human GM12878 cells. (**A**) APA plot of significant interactions detected by ChIA-PET Tool V3. (**B**) APA plot of significant interactions detected by ChIA-PET2. Significant interactions require at least three supportive PETs and FDR of less than 0.05. Interaction matrices are built with BEDPE files from ChIA-PET data after removing duplicates. Interaction counts are summed for all pairs of loci in 25 kb bins.

**Table 1 genes-10-00554-t001:** Statistics of linker composition.

	A_A	A_B	B_A	B_B	Ambiguous	Total
Numbers	33,029,693	420,122	425,932	35,544,028	6,155,769	75,575,544
Percentage	43.70%	0.56%	0.56%	47.03%	8.15%	100%

A_A: refers to PETs that both reads optimally aligned to linker A; A_B: refers to PETs that read1 and read2 optimally aligned to linker A and linker B, respectively; B_A: refers to PETs that read1 and read2 optimally aligned to linker B and linker A, respectively; B_B: refers to PETs that both reads optimally aligned to linker B; Ambiguous: refers to PETs that not satisfy either one of the criteria below: (1) best linker alignment score is more than the cutoff, (2) the difference between second-best and best linker alignment score exceeds score cutoff, (3) tag length should conform to the specified range, and (4) the barcodes must be completely matched with reads; Total: refers to total number of PETs.

**Table 2 genes-10-00554-t002:** Statistics of interactions with Paired-End Tag (PET) counts.

PET Counts	No. of Clusters	No. Intra-Chrom Clusters	No. Inter-Chrom Clusters	Percentage of Intra-ChromClusters
2	22,545	22,240	305	98.65%
3	4053	4035	18	99.56%
4	1437	1432	5	99.65%
5	749	745	4	99.47%
6	453	448	5	98.9%
7	298	296	2	99.33%
8	238	236	2	99.16%
9	173	173	0	100%
≥10	1065	1059	6	99.44%
Total	31,011	30,664	347	98.88%

**Table 3 genes-10-00554-t003:** Span distribution of interactions.

Distance	Frequency	Interaction Type
<100 kb	24,656	Intra-chromosomal
[100 kb, 1Mb]	5830	Intra-chromosomal
[1Mb, 10Mb]	131	Intra-chromosomal
>10Mb	47	Intra-chromosomal
Different chromosomes	347	Inter-chromosomal

**Table 4 genes-10-00554-t004:** Statistics of RNAPII ChIA-PET data from human K562 cells.

Category	Number	Percentage	Percentage of Total PETs	Order
Total PETs	75,575,544	N/A	N/A	(1)
Same-linker PETs after linker filtering	68,573,721	90.74% of (1)	90.74% of (1)	(2)
Uniquely mapped same-linker PETs	10,732,248	15.65% of (2)	14.2% of (1)	(3)
Merging same same-linker PETs	7,407,560	69.02% of (3)	9.8% of (1)	(4)
Merging similar same-linker PETs	7,388,164	99.74% of (4)	9.78% of (1)	(5)
Self-ligation PETs	2,387,126	32.31% of (5)	3.16% of (1)	(6)
Inter-ligation PETs	2,976,285	40.28% of (5)	3.94% of (1)	(7)
Other PETs with short distance	2,024,753	27.41% of (5)	2.68% of (1)	(8)
Peaks from self-ligation	6332	N/A	N/A	(9)
Interacting pairs	31,011	N/A	N/A	(10)

**Table 5 genes-10-00554-t005:** Statistics of ChIA-PET library from human K562 cells.

Category	Amount
Percentage of same-linker PETs over total PETs	90.74%
Percentage of uniquely mappable PETs over total PETs	14.2%
Percentage of PETs after merging those mapped to the same positions exactly over uniquely mappable PETs	69.02%
Percentage of inter-ligation PETs over PETs after purification	40.28%
Percentage of intra-chromosomal inter-ligation PETs over inter-ligation PETs	53.94%
Number of peaks	6332
Number of interactions	31,011

**Table 6 genes-10-00554-t006:** Statistics of CCCTC binding factor (CTCF) ChIA-PET data from human GM12878 cells.

Category	Number	Percentage	Percentage of Total PETs	Order
Total PETs	681,535,975	N/A	N/A	(1)
Same-linker PETs after linker filtering	350,456,404	51.42% of (1)	51.42% of (1)	(2)
Uniquely mapped same-linker PETs	256,270,827	73.12% of (2)	37.6% of (1)	(3)
Merging identical same-linker PETs	84,843,463	33.11% of (3)	12.45% of (1)	(4)
Merging similar same-linker PETs	81,226,332	95.74% of (4)	11.92% of (1)	(5)
Self-ligation PETs	26,630,193	32.79% of (5)	3.91% of (1)	(6)
Inter-ligation PETs	52,393,582	64.5% of (5)	7.69% of (1)	(7)
Other PETs with short distance	2,202,557	2.71% of (5)	0.32% of (1)	(8)
Peaks from self-ligation	29,829	N/A	N/A	(9)
Interacting pairs	1,762,737	N/A	N/A	(10)

**Table 7 genes-10-00554-t007:** Statistics of ChIA-PET library from human GM12878 cells.

Category	Amount
Percentage of same-linker PETs over total PETs	51.42%
Percentage of uniquely mappable PETs over total PETs	37.6%
Percentage of PETs after merging those mapped to the same positions exactly over uniquely mappable PETs	33.11%
Percentage of inter-ligation PETs over PETs after purification	64.5%
Percentage of intra-chromosomal inter-ligation PETs over inter-ligation PETs	30.49%
Number of peaks	29,829
Number of interactions	1,762,737

**Table 8 genes-10-00554-t008:** Statistics of different tools from human K562 cells.

Tools	Non-Chimeric PETs	Non-Duplicate PETs	Peaks	Significant Interactions (PET ≥ 3)
Mango	67,330,362	40,170,789	30,343	1676
ChIA-PET2	67,347,877	7,303,170	17,121	5179
ChIA-PET Tool V3	68,573,721	7,388,164	6332	8466
ChiaSig	N/A	N/A	N/A	7370

**Table 9 genes-10-00554-t009:** Statistics of different tools from human GM12878 cells.

Tools	Non-Chimeric PETs	Non-Duplicate PETs	Peaks	Significant Interactions (PET ≥ 3)
ChIA-PET Tool V3	350,456,404	81,226,332	29,829	420,494
ChIA-PET2	106,975,449	32,278,242	47,725	37,313

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
