# Peer review of "Chromatin Interaction Analysis with Updated ChIA-PET Tool (V3)"

_genes, 2019, doi:10.3390/genes10070554_

Round 1
The very initial submission was withdrawn; the current round was started with previous comments and reply provided to previous reviewers.
Reviewer 1 Report
1. With the addition of section 2.2, the description of the updates in ChIA-PET tool V3 is now sufficient. I must add, however, that the extent of the update in ChIA-PET Tool V3 (relative to previous versions) is not substantial, and largely involves re-implementation of code to increase computational speed/efficiency and user-friendliness.
2. The authors have NOT sufficiently answered my concerns regarding the description of how the significant interactions are identified, as they have not updated the manuscript with more information on this.
The included figure does clearly show that the genomic distances of the interactions specific to ChIA-PET Tool V3 (red) are shorter in genomic length. This fact should be mentioned in the manuscript, since this is relevant information for potential users of the tool. I was also puzzled by the small peak at log10(distance)>9. This means that ChIA-PET tool V3 reports interactions between anchors that are > 1 billion basepairs from each other. Surely, these must be artifacts.
3. This is now ok.
4. As mentioned in point 2 above, I suggest adding this information to the manuscript. This is important information for potential users of the tool.
5. I have tried running both the short- and long read test data using the commands provided on the GitHub.
In both cases, I get the following error:
Exception in thread "main" java.lang.NullPointerException
at process.Path.setParameter(Path.java:99)
at process.Main.main(Main.java:52)
I therefore still have not managed to run the software successfully.
Author Response
We are thankful for the reviewer’s comments. Please find our reply in the attachment.

Reviewer 2 Report
The authors addressed the comments well.
Author Response
We are thankful for the reviewer’s comments.